# Controlled gene and drug release from a liposomal delivery platform triggered by X-ray radiation

Wei Deng[1,7], Wenjie Chen[1], Sandhya Clement[1,7], Anna Guller [1,2,3,7], Zhenjun Zhao[2], Alexander Engel[4,5,6] & Ewa M. Goldys[1,7]

Liposomes have been well established as an effective drug delivery system, due to simplicity of their preparation and unique characteristics. However conventional liposomes are unsuitable for the on-demand content release, which limits their therapeutic utility. Here we report X-ray-triggerable liposomes incorporating gold nanoparticles and photosensitizer verteporfin. The 6 MeV X-ray radiation induces verteporfin to produce singlet oxygen, which destabilises the liposomal membrane and causes the release of cargos from the liposomal cavity. This triggering strategy is demonstrated by the efficiency of gene silencing in vitro and increased effectiveness of chemotherapy in vivo. Our work indicates the feasibility of a combinatorial treatment and possible synergistic effects in the course of standard radiotherapy combined with chemotherapy delivered via X-ray-triggered liposomes. Importantly, our X-ray-mediated liposome release strategy offers prospects for deep tissue photodynamic therapy, by removing its depth limitation.

[1] ARC Centre of Excellence for Nanoscale Biophotonics, Faculty of Science and Engineering, Macquarie University, North Ryde, 2109 New South Wales, Australia. [2] Faculty of Medicine and Health Sciences, Macquarie University, North Ryde, 2109 NSW, Australia. [3] Sechenov University, Moscow, 119992, Russia. [4] Sydney Medical School, University of Sydney, Sydney, 2050 NSW, Australia. [5] Department of Colorectal Surgery, Royal North Shore Hospital, St Leonards, 2065 NSW, Australia. [6] Sydney Vital Translational Cancer Research, Kolling Institute of Medical Research, Northern Sydney Local Health District, St Leonards, 2065 NSW, Australia. [7] Present address: The Graduate School of Biomedical Engineering, University of New South Wales, Sydney, Kensington, 2052 NSW, Australia. These authors contributed equally: Wenjie Chen, Sandhya Clement. Correspondence and requests for materials should be addressed to W.D. (email: wei.deng@mq.edu.au or (email: wei.deng@unsw.edu.au) or to E.M.G. (email: e.goldys@unsw.edu.au)

The development and application of various nanomaterial designs for gene and drug delivery is currently one of the key focus areas in nanomedicine. Although viral carriers have been traditionally used as a gene/drug delivery method[1,2], their application is hindered by a range of limitations including immunogenicity, limited size of transgenic materials, packaging difficulties and the risk of recombination[3]. Furthermore, viral carriers do not offer any temporal control over transfection which, once introduced, cannot be deliberately stopped[4]. To overcome these limitations, synthetic nanomaterial-based systems have been extensively studied and developed. Among these nanomaterials, liposomes have been well established as an effective drug delivery system, due to simplicity of their preparation and unique characteristics[5,6]. Liposomes consist of an aqueous core surrounded by a lipid bilayer similar to cell membranes, which facilitates cellular uptake of liposomes. The lipids forming liposomes are amphipathic, thus allowing the encapsulation of both hydrophobic and hydrophilic molecules or (and) colloidal particles[7]. Liposomes are usually biocompatible and biodegradable, which makes them suitable for clinical applications[5,8].

However conventional liposomes, for example, commercial lipofectamine 2000, are unsuitable for the on-demand content release, which limits their therapeutic utility, although they possess high efficiency of delivery. By contrast, triggerable liposomes are able to release genes/drugs in a more controlled manner, usually much faster and, depending on triggering modality, also to a specific area, and these properties contribute to their potentially greater clinical success. Several strategies have been previously employed to design responsive liposomes whose bilayer could be destabilised by using physiological and external stimuli. The triggering approaches previously reported include changes in pH (typical in cancer)[9,10], externally delivered heat, for example via alternating magnetic field or infrared light[11,12], enzymes[13,14] and non-thermal effects caused by light irradiation[15,16]. These approaches have certain limitations, in particular triggering of light-sensitive liposomes by visible light is limited by its relatively shallow (few mm) penetration of light into biological tissues[17]. As a result of this modest penetration depth, visible light can not activate photosensitizers (PS) located deeply in the body and generate sufficient amount of singlet oxygen ($^1O_2$) or other reactive oxygen species (ROS) to release the liposome cargo required for the therapeutic effects[18]. With its excellent tissue penetration depth, X-ray radiation explored in this work for liposome triggering offers an alternative approach to yield both spatial targeting (such as to a tumour site) via standard radiotherapy approaches such as the Gamma-knife[19] and triggered release of encapsulated contents from the liposomes once they are located at the target site. Importantly, the X-ray liposome triggering can be used concurrently with radiation therapy, a common treatment modality in cancer.

Herein we design X-ray triggered liposomes by co-embedding photosensitizers and gold nanoparticles (3–5 nm) inside a lipid bilayer. Gold is chosen in this work as, due to its high atomic number, it strongly interacts with X-ray radiation as shown, for example, by gold nanoparticle-induced radiation enhancement inside biological tissue[20–22]. Although in our design the photosensitisers are the primary source of reactive oxygen species (ROS) to oxidise unsaturated lipids and destabilise liposomal membranes, gold nanoparticles exposed to X-rays also generate some level of ROS[23]. More complex effects are also possible; for example, secondary electrons produced during the interaction of X-rays with gold nanoparticles may transfer from gold to a photosensitizer and lead to PS-induced generation of $^1O_2$ or other ROS[24–26]. As a photosensitizer we choose verteporfin (VP), clinically approved for photodynamic therapy (PDT) of age-related macular degeneration[27,28]. 1,2-dioleoyl-sn-glycero-3-

phosphocholine (DOPC) and 1, 2-di-(9Z-octadecenoyl)-3-trimethylammonium-propane (DOTAP) are chosen as lipid components in the liposome formulation because DOPC can load highly hydrophobic molecules and DOTAP can facilitate cellular uptake due to its positive charge[29]. The $^1O_2$ generation from different liposome samples and destabilization of the lipid bilayer by $^1O_2$ under 365 nm LED illumination with different time points (2, 4, 6, 8 and 10 min) and X-ray radiation with different dosage (1, 2 and 4 Gy) are assessed by using singlet oxygen green sensor (SOSG) and calcein release assays, respectively. SOSG is a commonly used and highly specific fluorescence probe for the detection of $^1O_2$ generation[30]. It is identified to be fluorescein covalently bound with an anthracene moiety[31]. Calcein is a fluorescent dye that self-quenches at high concentration[32,33] which makes it possible to detect its release from the liposomes to the surrounding environment by monitoring the increase in calcein fluorescence intensity upon X-ray radiation[34,35]. Additionally, $^1O_2$ quantum yield under UV light illumination and the number of $^1O_2$ generated as a result of X-ray radiation are also calculated based on experimental data[36,37]. Triggered release of the liposome cargo by X-rays is verified by (a) demonstrating the efficiency of X-ray triggered gene silencing in vitro and (b) the increased effectiveness of chemotherapy in vivo (Fig.1). For gene silencing, an antisense oligonucleotide complementary to a specific pituitary adenylate cyclase-activating polypeptide (PACAP) receptor, PAC1R, is encapsulated inside the liposomes. Following the liposome take-up by rat PC12 cells, the X-ray radiation at a dose of 4 Gy is applied. As a result of exposure to ionising radiation, the $^1O_2$ generated in a lipid bilayer destabilises the liposomes, leading to the release of antisense oligonucleotides. This antisense nucleotide is then able to prevent the translation of the PAC1R mRNA by blocking the translation initiation complex. Gene knockdown is monitored by observing a decrease in the fluorescence intensity from indirect immunofluorescence staining

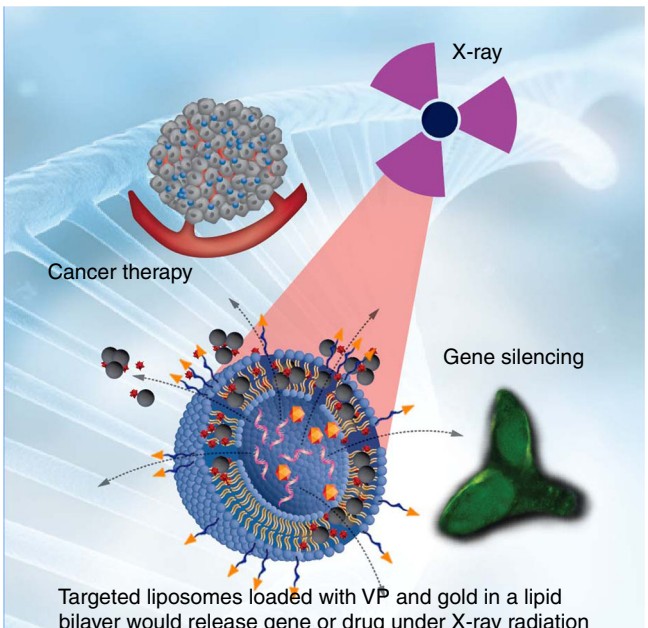

**Fig. 1** The schematic illustration of gene silencing and cancer cell-killing by X-ray-triggered liposomes. This liposomal delivery platform incorporates verteporfin (VP) and gold nanoparticles. Two types of cargos, antisense oligonucleotide and Doxorubicin, are respectively entrapped inside a liposomal middle cavity for demonstration of in vitro gene release and in vivo drug delivery

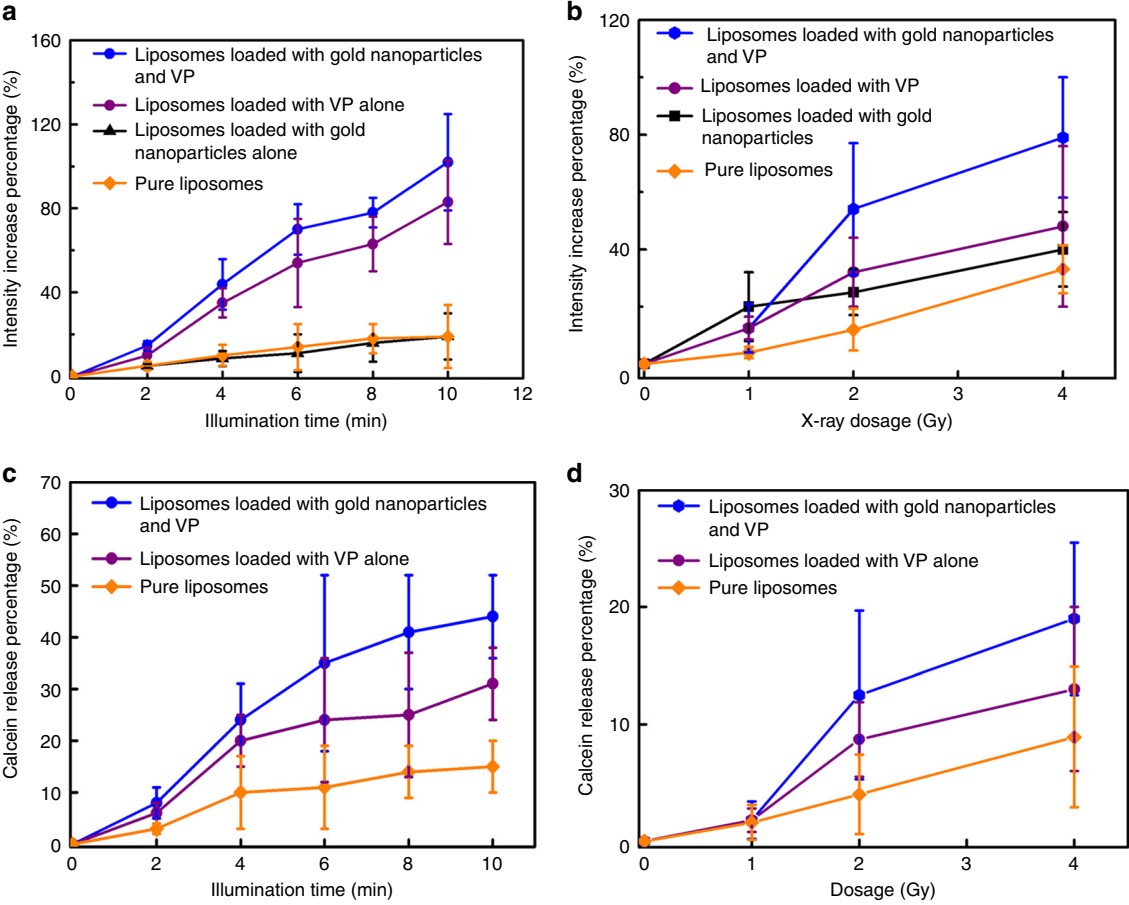

**Fig. 2** Singlet oxygen generation and calcein release from liposomes under light and X-ray triggering. **a**, **b** Percentage increase of SOSG fluorescence intensities from different liposome samples under (**a**) 360 nm irradiation at different time points and (**b**) X-ray radiation with different doses. **c**, **d** Calcein release profiles from liposomes under (**c**) 360 nm irradiation and (**d**) X-ray radiation. Error bars show standard deviation from four measurements

of PAC1R in cells after X-ray irradiation. For X-ray-triggered chemotherapy, an antitumour drug, doxorubicin (Dox), is loaded into the liposomes. The liposomes are taken up by human colorectal cancer HCT 116 cells and X-rays applied. In vivo antitumour effect is evaluated by monitoring tumour development and body weight of mice bearing colorectal cancer xenografts and by conducting histological analysis of tumour tissues after the treatments.

## Results

**$^1O_2$ generation tests by using light and X-rays respectively**. The generation of singlet oxygen is a key factor in the oxidation of unsaturated lipids, resulting in the disruption of the liposome structure[26]. $^1O_2$ generation was confirmed by using SOSG and monitoring the enhancement of fluorescence intensity at 488 nm excitation. $^1O_2$ reacts with SOSG to produce endoperoxides which have a strong fluorescence signal at 525 nm for 488 nm excitation, while it has weak fluorescence in the absence of $^1O_2$. The SOSG fluorescence intensity enhancement as a function of light illumination time and X-ray dose, respectively, is plotted in Fig. 2. Figure 2a shows that the liposomes loaded with gold nanoparticles and VP generate more singlet oxygen than the other samples, with an increase of about 102% after 10 min illumination. Singlet oxygen quantum yield (SOQY) from this sample (liposomes loaded with gold nanoparticles and VP) is calculated to be $0.75 \pm 0.18$ (mean value ± standard deviation), indicating an enhancement factor of 1.42 compared with the

liposomes loaded with VP alone. The details of this calculation are explained in Supplementary Note 2. We attribute this enhancement of $^1O_2$ generation from VP to near-field enhancement of electromagnetic field induced by gold nanoparticles[38,39]. However such enhancement was dependent on one of experimental factors, the distance between gold and photosensitisers. In this study the distance between gold nanoparticles and VP molecules was not controllable under the current condition because both were randomly loaded in the liposomal bilayer, with some molecules less than optimally placed in terms of the distance for optimal enhancement of singlet oxygen generation. This may partially contribute to the singlet oxygen generation enhancement limited within a certain range as observed in this study. In particular, the interaction between gold and photosensitisers would not contribute to the singlet oxygen generation when they are extremely close[40,41].

Similarly, the enhancement of $^1O_2$ generation was observed in liposomes loaded with gold nanoparticles and VP in our X-ray radiation experiments as well, but to a lesser extent. As shown in Fig. 2b, liposomes doped with gold nanoparticles and VP molecules generate the highest amount of $^1O_2$, with a percentage increase of approximately 79% under X-ray radiation with 4 Gy, while liposomes containing gold nanoparticles alone and the sample containing VP alone produced a limited amount of $^1O_2$, with a percentage increase of approximately 48% and 40%, respectively, under the same experimental conditions. We calculated the number of singlet oxygen generated from liposomes loaded with VP and gold nanoparticles under X-ray

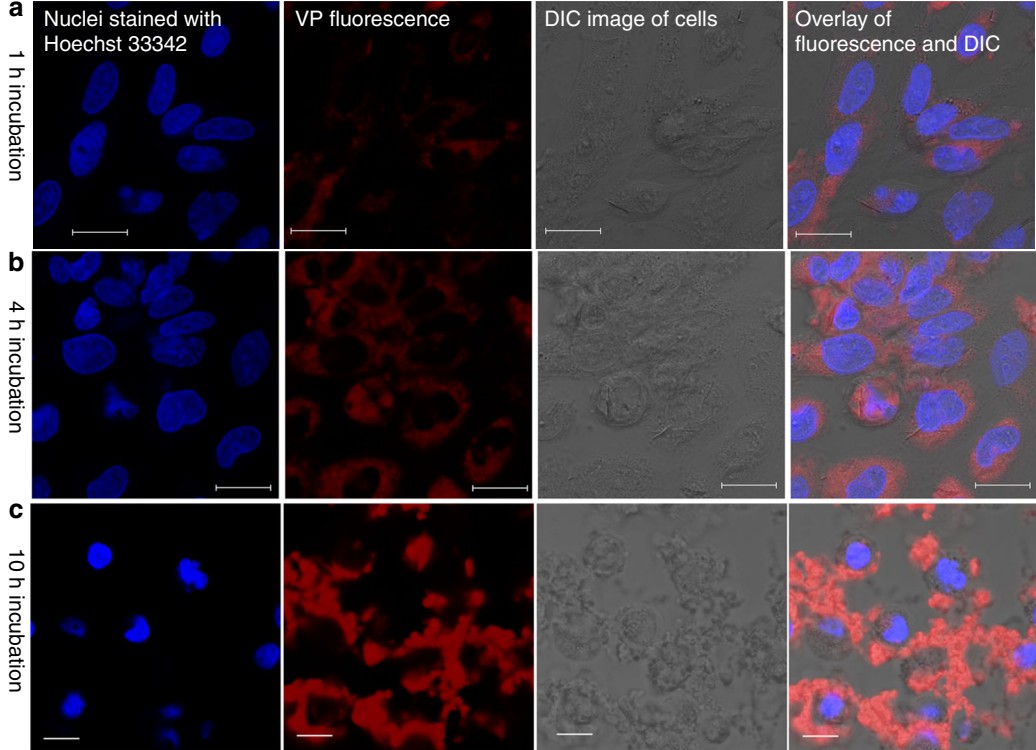

**Fig. 3** Cellular uptake activity of liposomes in rat PC12 cells. **a–c** Representative confocal laser scanning microscopy images of PC12 cells incubated with liposome nanoparticles (25 μM) for 1, 4 and 10 h, respectively. Scale bar is 20 μm

radiation with 4 Gy, to be 7250 per a single liposome. The calculation is provided in Supplementary Note 3. The observed enhancement of X-ray induced $^1O_2$ generation in the presence of gold nanoparticles can be explained by the following mechanism. Gold is a heavy metal element strongly interacting with X-rays, which leads to a significant increase of energy deposition in biological tissues when irradiated with such rays[42–44]. Therefore gold nanoparticles are well-known radiosensitizers able to amplify the radiation doses in tumour tissue[45–47]. In addition, gold nanoparticles can selectively scatter and (or) absorb the high energy X-ray radiation[20–22], leading to enhanced energy transfer from X-ray to photosensitizers. With such contribution, the VP molecules in the presence of gold nanoparticles are able to interact more strongly with ionising radiation than the VP on its own, causing enhanced $^1O_2$ generation.

**Calcein release assays under two external stimuli**. Having confirmed the $^1O_2$ generation from VP entrapped inside liposomes using two stimulating modalities, we attempted to evaluate the liposome content release by using a calcein release assay, which is based on the principle of fluorescence self-quenching[48,49]. Figure 2c, d shows the proportion of calcein release from different liposome samples under UV illumination and X-ray exposure, respectively. The amount of calcein released from liposomes doped with both gold nanoparticles and VP reaches a maximum of 44% after 10 min light illumination (Fig. 2c) and 19% after X-ray radiation with 4 Gy (Fig. 2d), respectively. However, lower leakage is observed in the controls (liposomes doped with VP alone), with only 31 and 13% of calcein being released at the same experimental conditions. Similarly to our results of the $^1O_2$ generation, our findings show that introduction of gold nanoparticles inside liposomes contributes to increased release of entrapped calcein, compared with samples

containing VP molecules only, under both UV illumination and X-ray radiation.

**Cellular uptake of liposomes in PC12 cells**. In order to investigate the cellular uptake of liposomes, the PC12 cells were treated with liposomes for 1 h, 4 h and 10 h. As shown in Fig. 3, higher red fluorescence signal from VP was observed after 4-h incubation compared with cells treated for 1 h. Detailed characterisation of the cellular uptake of liposomes after 4 h incubation with PC12 cells is provided in the Supplementary Fig. 6. In addition, green fluorescence from fluorescein amidite (FAM)-labelled oligonucleotide is also clearly observed after 4 h incubation (Supplementary Fig. 7). After 10 h incubation with liposomes, cells were surrounded by large red clusters, indicating a large amount of liposomes loaded with VP were internalised by cells. However, some clusters were also observed in other regions due to non-specific binding (Fig. 3). Therefore, we chose 4 h incubation time for PC12 cells. Based on the concentration of fluorescently labelled lipid internalised by cells, we estimated that 2550 ± 89 liposomes were internalised by each PC12 cell. The number of gold nanoparticles per liposome is estimated to be 156 ± 24 on the basis of the ICP-MS data. Therefore, the number of gold nanoparticles internalised by each PC12 cell is estimated to be 3.98 × $10^5$ in this study. The detailed calculation of the number of liposome per cell and the number of gold nanoparticles per liposome is provided in Supplementary Note 4.

**Cellular uptake of folate-conjugated liposomes**. The folate receptor (FR) is significantly expressed in many types of cancer cells while its expression in most normal tissues is generally low[50]. Folic acid (FA) has a very high affinity for the FRs with a minimal effect on its binding ability even after conjugation with other nanomaterials. Therefore FA can significantly enhance the

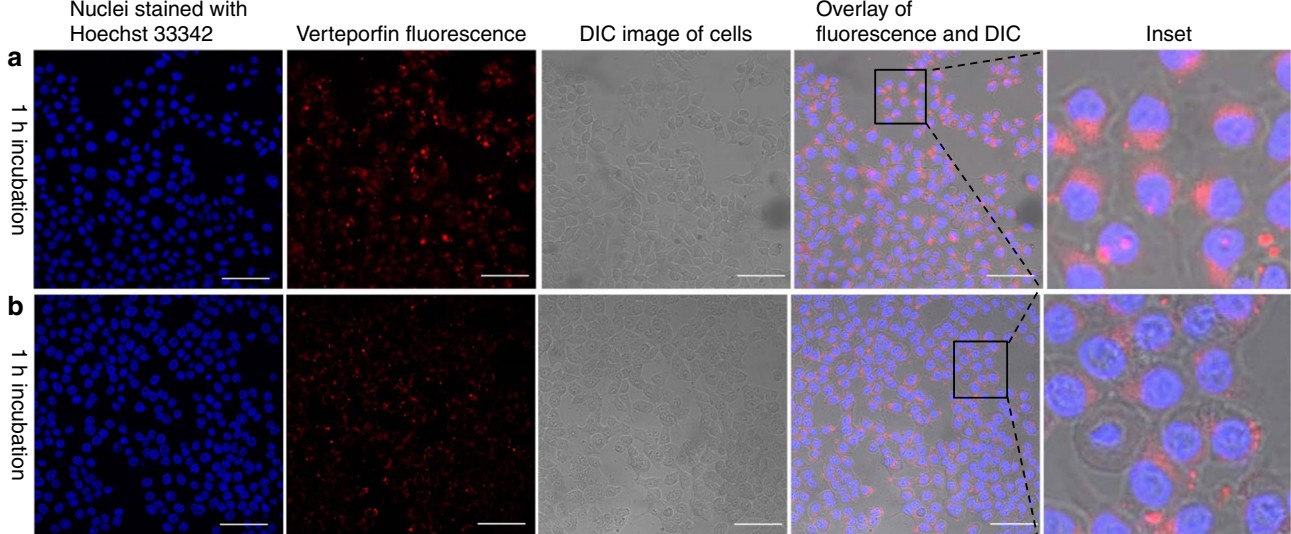

**Fig. 4** Cellular uptake of folate-conjugated liposomes in HCT 116 cells and CCD 841 CoN cells. **a**, **b** Representative confocal laser scanning microscopy images of incubated (**a**) HCT 116 cells and (**b**) CCD 841 CoN cells with folate-conjugate liposomes (25 μM) for 1 h. Scale bar is 75 μm

capability of nanoparticle-based delivery systems to target cancer cells[51,52]. In this study, we modified the liposome surface with folate and determined the average number of the folate molecules per liposome (estimated to be approximately 480) based on the total amount of folate and liposomes in the sample. To evaluate the targeting specificity of the folate-targeted liposomes to tumour cells, the uptake activity of liposomes by colorectal cancer HCT 116 cells, was compared to the uptake by normal human colonic cell line, CCD 841 CoN. As shown in Fig. 4a, cancer cells treated with folate-conjugated liposome nanoparticles clearly exhibited red signal from VP in the cytoplasm after 1 h incubation. By contrast, the level of liposome uptake by CCD 841 CoN cells is shown to be fairly low under the same experimental conditions (Fig. 4b). These results indicated that FA- induced specific binding to the folate receptor expressed on HCT 116 cell surface resulted in a higher internalisation rate of targeted liposomes, compared to the normal CCD 841 CoN cells.

**X-ray triggered in vitro gene silencing and chemotherapy**. We further applied the liposomes loaded with antisense oligonucleotide to carry out the PAC1R gene knockdown by delivering the liposomes to PC12 cells and applying 4 Gy of X-ray radiation. The fluorescently labelled PAC1R expressed by PC12 cells was imaged by using confocal microscopy at various time points. For comparison, the cells treated with liposomes alone, but without triggering were also imaged using the same imaging conditions. As shown in Fig. 5a, decreased fluorescence in cell samples was clearly observed 24 h after X-ray exposure, indicating that the antisense oligonucleotide released from liposomes effectively knocked down the PAC1R gene expression. For cells treated with liposomes alone, a decreased PAC1R fluorescence signal was also observed at 24 h after treatment, but the decrease was less pronounced compared to cells treated with X-ray radiation (Fig. 5b). We quantitatively analysed the PAC1R inhibition at different time points based on cellular fluorescence images. After 24 h since X-ray exposure the density of PAC1R decreased by about 45%, while the level of PAC1R in cells which were not exposed to X-rays but received the liposomes with antisense oligonucleotides decreased by only 30% (Fig. 5c, d).

In addition to the demonstration of gene silencing by using X-ray-triggered liposomes, we also investigated the in vitro cell-

killing effect of the liposomes loaded with varying amounts of antitumour drugs, Dox and etoposide (ETP), in HCT 116 cells. A series of drug-dilution assays presented in Supplementary Figure 8a reveals that 50% cell-killing (IC$_{50}$) was achieved at 1.6 μM of Dox encapsulated in the liposomes (LipoDox) and triggered by X-ray radiation. However, the LipoDox alone, without X-ray triggering but with same Dox concentration of 1.6 μM killed only about 10% of cancer cells (Supplementary Fig. 8b). This illustrates, not unexpectedly, that the efficacy of LipoDox for cell killing was higher with X-ray radiation, compared with LipoDox only. The results of our X-ray-triggered LipoDox treatment described here indicates that a combination of X-ray-triggered chemo- and radiotherapy with the same X-rays appears to produce an enhanced effect and it yields improved efficacy of cancer cell-killing. It should be mentioned that simultaneous chemo- and radiotherapy may result in the development of cardiotoxicity, whose incidence is associated with different factors, including the type of antitumour drugs[53]. Therefore, we evaluated the cell-killing effect of a second chemotherapy drug, ETP, in combination with X-ray radiation. ETP is associated with reduced incidence of cardiotoxicity, compared with Dox[54]. As shown in Supplementary Fig. 8c, higher cytotoxicity of LipoETP in HCT 116 cells was observed at 24 h after X-ray radiation of 4 Gy, compared with LipoETP alone.

**Toxicity assays of liposomes and X-ray exposure**. We first assessed the toxicity of liposomes doped with gold nanoparticles and VP. Compared with the control group, no significant change was observed in the viability of PC12 cells treated with liposome concentrations up to 50 μM, higher than those used for gene and drug delivery in our study (Supplementary Fig. 9a). The liposome-formulated Dox designed in this study should also have minimal toxicity effect on normal cells without X-ray-triggering. To verify this, we examined the toxicity of LipoDox on CCD 841 CoN cells by varying Dox concentration. As shown in Supplementary Fig. 9b, we did not observe a noticeable reduction in cell survival (up to 14% cell death) at 24 h after incubation with liposome-formulated Dox samples (Dox concentration: 3 μg mL$^{-1}$ and 2 μg mL$^{-1}$), suggesting that under in vitro conditions, our LipoDox samples with these two Dox concentrations are likely not to affect the viability of CCD 841 CoN cells.

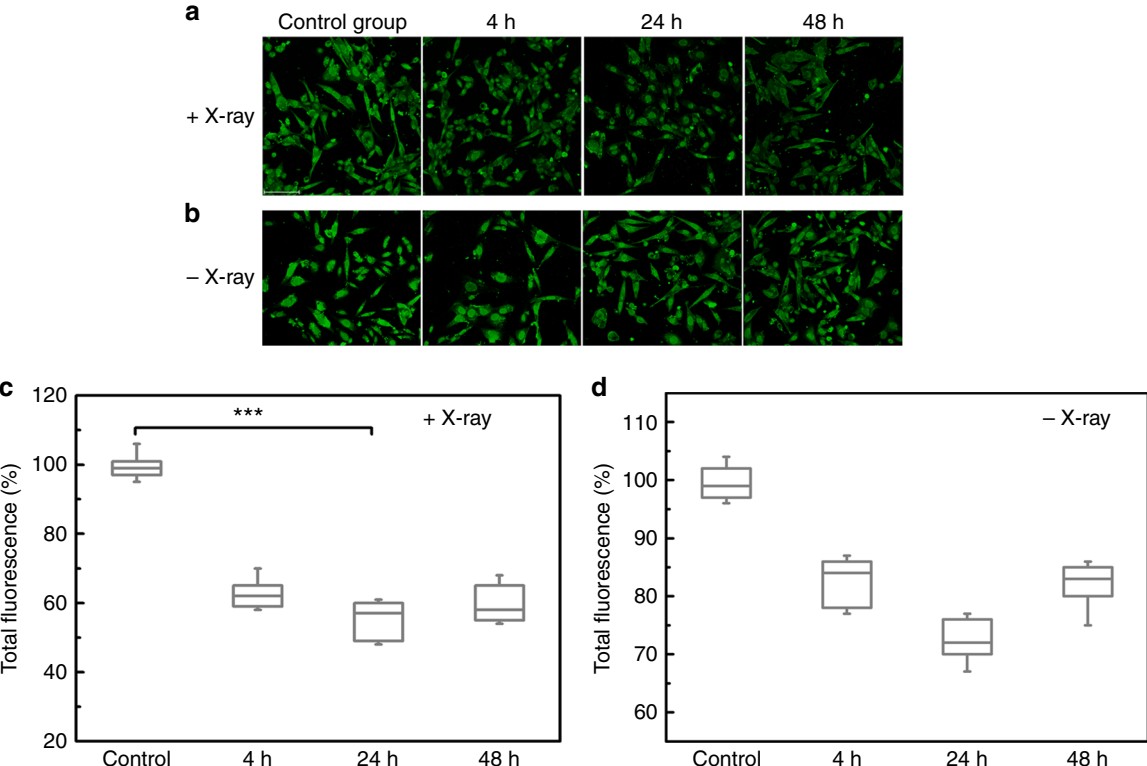

**Fig. 5** In vitro gene silencing by X-ray triggered liposomes loaded with antisense oligonucleotide. **a**, **b** Representative confocal images of indirect immunofluorescence staining of PAC1R at different time points after cells were treated with (**a**) X-ray-triggered liposomes and (**b**) liposomes alone. The concentration of liposomes incubated with cells was 25 µM. Scale bar was 75 µm. Boxplots in **c**, **d** show quantitative assessment of PAC1R gene silencing induced by antisense oligonucleotide released from liposomes at different time points (**c**) with and (**d**) without X-ray radiation. Decreased PAC1R fluorescence intensity was expressed as percentage of the control. The box is bounded by the first and third quartile with a horizontal line at the median and whiskers extend to 1.5 times the interquartile range. The mean value was analysed using the $t$ test ($n = 5$). *** $P < 0.001$, compared with the control group

It is well known that radiolysis of water molecules as a result of X-ray radiation damages DNA molecules by producing toxic radicals. Although cells attempt to repair the damage, complete repair may not be possible at higher doses[55]. The surviving cells may suffer residual DNA damage, potentially contributing to adverse long-term health effects. In this study we assessed the X-ray-induced damage in both cultured cells and genetic materials. In cell experiments, the MTS test did not reveal a clear decrease in survival of PC12 cells, HCT 116 cells and CCD 841 CoN cells at 24 h and 48 h after X-ray exposure (Supplementary Fig. 9c). With regard to the X-ray effects on genes, the DNA gel electrophoresis did not show obvious dispersion of DNA bands after X-ray radiation compared to the control, indicating that X-ray radiation with our applied dosage did not cause obvious damage to the DNA molecules (Supplementary Fig. 9d).

In addition, we also checked the effect of the singlet oxygen on genetic material by irradiating a mixture solution of oligonucleotides and VP with X-ray. As shown in Supplementary Fig. 9d, there was no clear oligonucleotide damage observed compared with the control. Singlet oxygen is the primary cytotoxic agent responsible for photobiological activity involved in the PDT technique. It can damage cells by reacting with many biomolecules, including amino acids, nucleic acids and unsaturated fatty acids that have double bonds as well as sulphur-containing amino acids[56,57]. Short lifetime of singlet oxygen prevents it from travelling larger distances, therefore it mainly causes localised[58], near the photosensitizer molecule where it was generated. In this study, singlet oxygen generated from VP loaded in a lipid bilayer mainly destabilises the unsaturated lipids and consequently induces drug release. This reaction with lipids consumes singlet oxygen radicals[59]. Therefore, adverse effect of singlet oxygen on oligonucleotides will be minimised.

**Therapeutic effect of X-ray-triggered liposomes in vivo**. To determine the efficacy of X-ray-triggered liposomes in vivo, we detected their ability to control tumour growth in a xenograft mouse model bearing HCT 1116 cells. Based on the in vitro work, 4 Gy was chosen for irradiation on mice. The sizes of tumours in mice treated with different conditions are presented in Fig. 6a. PBS-, liposome- and X-ray-treated tumours respectively increased 3.0-fold, 2.9-fold, and 3.4-fold during the study period (two weeks post treatment), indicating that these treatments failed to delay tumour progression. By contrast, in the group treated with X-ray-triggered liposomes the tumour sizes gradually shrunk over this period, with 74% reduction in tumour volume compared to the PBS control group. The size of tumours in mice exposed to different treatments were also photographed and presented in Supplementary Fig. 10a. This figure shows that tumours in mice treated with X-ray-triggered liposomes grew more slowly in comparison with PBS control, X-ray radiation alone, and liposomes alone. In addition, no mortality was observed during 14 days after treatment with X-ray-triggered liposomes, and no weight loss of treated mice was observed compared to the control, suggesting that this combined technique is well tolerated by mice under the present conditions (Fig. 6b). Histological analysis was also performed to further verify the tumour response to the

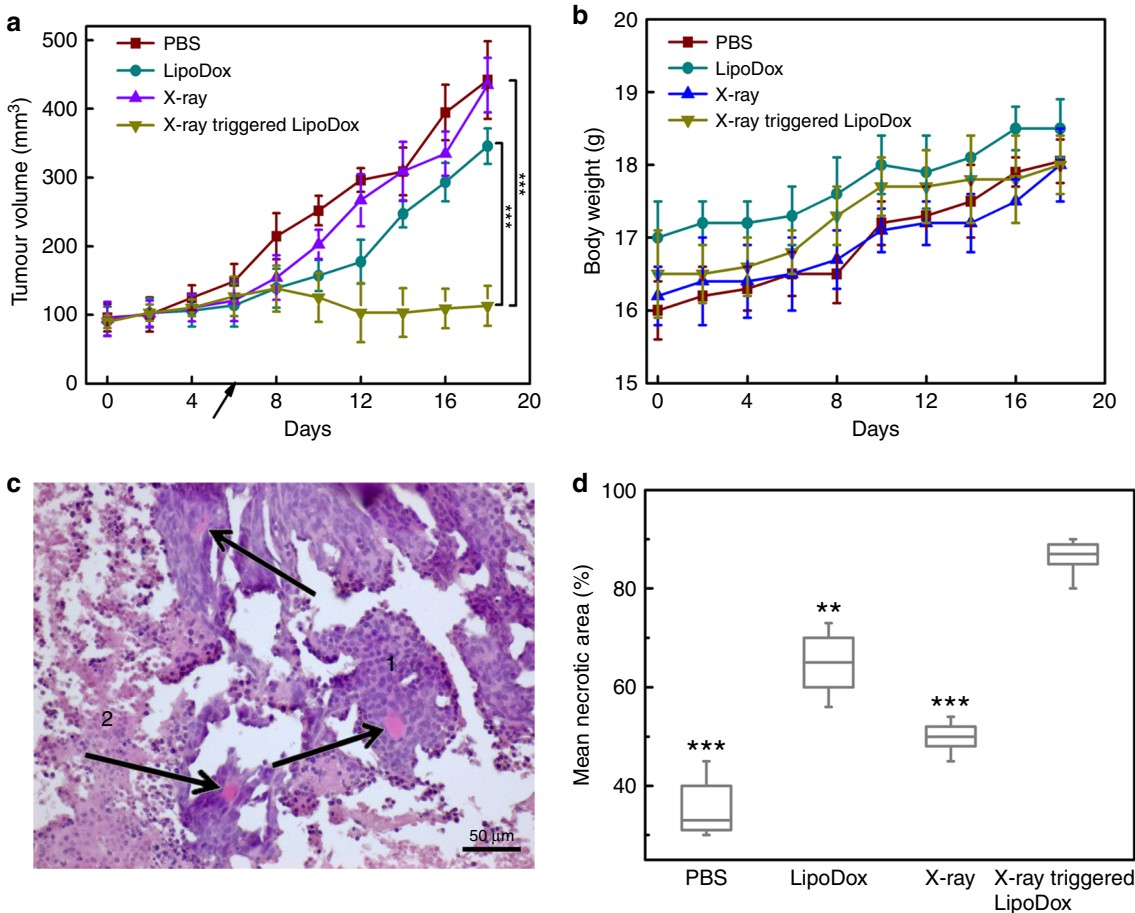

**Fig. 6** Antitumour activity of X-ray triggered LipoDox in a xenograft model of colorectal cancer. **a,b** Changes of (**a**) tumours' volume and (**b**) mouse body weight after various treatments as indicated. A black arrow indicates the time of treatment administration. Error bars show standard deviation from four experiments. The mean tumour volumes were analysed using the $t$ test ($n = 4$). * $P < 0.05$, ** $P < 0.01$, *** $P < 0.001$. **c** The structural components of treated tumour (H&E staining). Viable tumour tissues (1) were composed of uniform cells with basophilic (blue) cytoplasm and large roundish hyperchromatic nuclei. The areas of cellular paranecrosis and necrosis (2) were recognised by disorganised groups of tumour cells with eosinophilic (pink) cytoplasm, with and without nuclei, respectively. Arrows indicate congested blood vessels. Note the spatial association between the viable tumour tissue and blood vessels. Scale bar is 50 μm. **d** Boxplot shows morphometric analysis of the effect of the experimental treatment regimens on the structural composition of the xenograft tumours. The relative areas of the viable and non-viable (paranecrotic and necrotic) tumour tissues were measured using ImageJ open source software. The box is bounded by the first and third quartile with a horizontal line at the median and whiskers extend to 1.5 times the interquartile range. The mean tumour necrosis percentage was analysed using the $t$ test ($n = 5$). * $P < 0.05$, ** $P < 0.01$, *** $P < 0.001$, compared with X-ray-triggered LipoDox-treated group

treatments. All tumours were found to be localised sub-cutaneously and surrounded by a thin capsule of connective tissue. No tumour invasion into the capsule tissue was observed. The tumours had a mixed histological structure, with various spatial combinations of viable, paranecrotic and necrotic tumour tissues (Fig. 6c). In general, viable tissues were localized mainly at the periphery of the tumours or near the blood vessels, while the non-viable elements were found more centrally, implying the contribution of intrinsic tumour hypoxia and the oxidative stress induced by the experimental treatments to the suppression of tumour growth. The mean percentage of necrotised tumour tissues showed statistically significant differences between the studied groups, with the maximal tumour necrosis being achieved when treated with X-ray triggered LipoDox (Fig. 6d). These findings further confirmed that this strategy can achieve better therapeutic effect compared with individual modality treatment. More detailed histological analysis for each treatment (PBS-, X-ray-, LipoDox-, and X-ray-triggered LipoDox-treated) is provided in Supplementary Note 6.

## Discussion

X-ray radiation, as an external liposome triggering modality, was employed to activate a liposomal gene/drug delivery system in this study. Our X-ray- triggerable liposomes were designed by encapsulating a photosensitizer, VP, and gold nanoparticles in a liposomal bilayer. When these liposomes were exposed to X-rays, enhanced $^1O_2$ generation from VP was achieved due to the interaction between gold nanoparticles with incident X-rays. This $^1O_2$ oxidises unsaturated lipids and destabilises the membrane, allowing the release of entrapped cargos from the liposomes. We demonstrated that this release strategy has the capacity for in vitro gene knockdown and enhanced cancer cell-killing efficacy by releasing two kinds of cargos, antisense oligonucleotide against PAC1R gene and an antitumour drug (Dox) upon X-ray radiation. In animal experiments, X-ray-triggered liposomes were demonstrated to control colorectal tumour growth more effectively than other individual modality treatment conditions.

X-rays and other forms of ionising radiation are used to diagnose and treat medical conditions and are known to

contribute to DNA mutations that may lead to dose-dependent and stochastic toxic effects. Compared with light, however, X-rays with the suitable energy can easily penetrate the human body, activating gene/drug release in deep tissues once the X-ray-triggered liposomes reach their target. This feature will open many opportunities for biomedical research and clinical medicine, from triggered gene therapies and chemotherapy, through to enhanced PDT which currently suffers from limited penetration depth of illumination light (usually in the UV and visible region).

Additionally, the strategy described here has been designed to be compatible with future clinical translation. The materials and approaches used in this study, such as VP, lipids, Dox, and X-rays, are clinically used in treatment of tumours. Although gold nanoparticles used in this study have not yet been approved by the regulatory agencies, their size is compatible with the requirements of renal clearance[60]. In this way, long-term nanoparticle toxicity is likely to be minimised if not eliminated. Moreover, the ease of conjugation of targeting ligands to liposome surface with appropriate linkers, for example, lipid-polyethylene glycol (PEG)[61], would be an added advantage when applied to the targeted therapy, in particular for tumour treatment. From a clinical point of view, it would be beneficial to have access to this multimodality treatment, given our evidence of better therapeutic effect (or, potentially, equal therapeutic effect) at diminished toxicity in the case when single modality treatment options alone can only produce desired therapeutic effects at a significant cost of short- and long-term toxicity.

## Methods

**Preparation of liposomes loaded with gold and VP**. 350 μL of DOTAP (Avanti Polar Lipids, no. 890890 P) dissolved in chloroform (100 mg mL$^{-1}$, Sigma-Aldrich, no. 288306-1 L) was mixed with 370 μL of DOPC (Avanti Polar Lipids, no. 850375 P) dissolved in chloroform (100 mg mL$^{-1}$), followed by addition of 40 μL of gold nanoparticle suspension (Nanocomposix, Inc) and 50 μL of VP (Sigma-Aldrich, no. SML0534-5MG) dissolved in dimethyl sulfoxide (DMSO, 2.3 mg mL$^{-1}$, Sigma-Aldrich, no. 472301-500 ML). For the synthesis of empty liposomes, VP and gold nanoparticles were omitted in the mixture solution. The mixture was diluted to 1.0 mL in total volume using chloroform and vortexed gently for 10 min. Chloroform was evaporated off with a stream of Argon and the remaining DMSO was evaporated under freeze-drying, which was carried out in a freeze dryer (Alpha 1–4 LDplus, John Morris Scientific Pty Ltd). The lipid film was hydrated by adding 1.0 mL of DI water to a glass test tube, followed by vigorous stirring until the suspension was homogenised. The hydrated lipid suspension was left overnight to allow the maximal swelling of liposomes. The suspension was then extruded eleven times in an extruder (Avanti Polar Lipids, Inc) with two 1.0 mL glass syringes. The pore size of the polycarbonate membrane (Avanti Polar Lipids, Inc) was 200 nm. The resulting suspension was stored at 4 °C under argon. For encapsulation of calcein inside liposomes, 1.0 mL calcein solution (100 mM, Sigma-Aldrich, no. C0875-5G) was used as lipid hydration solution, instead of DI water. For encapsulation of oligonucleotides, 1.0 mL PBS (pH 7.4) solution containing antisense oligonucleotide (10 μM, 5′-TGGTGCTTCCCAGCCACTAT-3′) with 3′ FAM labelling against PAC1R gene (Integrated DNA Technologies Pte. Ltd.) was used to hydrate lipid film, followed by the hydration procedure described above. In order to remove calcein and oligonucleotides present in the supernatant after hydration, liposomes were then centrifuged at 14000×$g$ for 10 min by using Pall Nanosep centrifugal devices (Sigma-Aldrich) as per manufacturer's instructions.

**Synthesis of LipoDox**. The encapsulation of doxorubicin inside of liposomes was conducted as per a published protocol, using a gradient exchange method with minor modifications[62]. 1 mL ammonium sulphate (250 mM, Sigma-Aldrich, no. A4418-100G) was added to the glass test tube where the lipid film was produced after evaporation of organic solvent, followed by the hydration procedure described above. Free ammonium sulphate was removed by dialysis in the PBS solution (pH 7.4) with buffer exchange repeated four times. The Dox solution (10 mg mL$^{-1}$, Sigma-Aldrich, no. D1515-10MG) was subsequently added to hydrated liposome suspension with a drug to lipid mass ratio of 1:10, followed by incubation at 60 °C for 1 h. Unloaded Dox was removed by dialysis in PBS solution (pH 7.4) with four time buffer exchange.

**Preparation of liposome incorporating ETP**. Liposomes incorporating ETP, VP and gold nanoparticles were prepared by thin film hydration with some modifications. Briefly, 100 μL of DOTAP (50 mg mL$^{-1}$ in chloroform) was mixed with 54 μL of DOPC (100 mg mL$^{-1}$ in chloroform), followed by addition of 6 μL of gold nanoparticle suspension, 7 μL of VP (2.3 mg mL$^{-1}$ in DMSO) and 83.5 μL of ETP (Sigma-Aldrich, no. E1383-25MG, 1 mg mL$^{-1}$ in chloroform and ethanol (1:1 V/ V)). After evaporation of organic solvent, the lipid film was hydrated with 1 mL PBS (pH 7.4). The hydration and extrusion procedure was the same as described above. The unloaded etoposide was removed by dialysis in the PBS solution (pH 7.4) with buffer exchange repeated four times.

**Preparation of folate-conjugated liposomes**. Folate-conjugated liposomes were prepared by postinsertion of DSPE-PEG2000-Folate micelles into preformed liposomes with slight modifications[63,64]. In brief, 1 mg DSPE-PEG2000-folate (Avanti Polar Lipids, no. 880124) was dissolved in 320 μL DMSO, followed by hydration with 3.1 mL of distilled water, producing 100 μM micelle suspension. The suspension was then dialysed three times in a 10000 MWCO dialysis tubing against 1 L water to remove DMSO. After this, 40 μL of micelles were added to 1 mL of the preformed liposome suspension in ammonium sulphate (250 mM) and heated at 60 °C for 1 h to produce folate-tethered liposomes. Leaked ammonium sulphate and unincorporated micelles were removed by dialysis. To determine the folate content conjugated with liposomes, bare liposomes was used in conjugation procedure instead of VP-loaded liposomes. After preparation, the folate amount was determined by measuring the UV absorbance at 285 nm after lysing liposomes with 0.1% Triton X-100 and comparing with a standard curve of folic acid with the known concentration.

**Characterisation of liposomes**. The extinction spectra of liposomes loaded with gold nanoparticles and VP, VP alone and gold nanoparticles alone were measured using a spectrophotometer (Cary 5000 UV-Vis-NIR, Varian Inc.). Size distribution and zeta potentials of liposomes were measured with a Zetasizer Nano Series from Malvern Instruments. The morphology of liposomes was documented using Transmission Electron Microscopy (TEM). For TEM imaging, the liposome samples were prepared by placing a drop of suspension onto a copper grid and air-dried, following negative staining with one drop of 2% aqueous Uranyl Acetate for contrast enhancement. The air-dried samples were then imaged using a PHILIPS CM 10 system at an accelerating voltage of 100 KV. Images were captured with an Olympus Megaview G10 camera and iTEM software. To determine the encapsulation efficiency of oligonucleotides, Dox and etoposide loaded inside of liposomes, Triton X-100 (0.1%, Sigma-Aldrich, no. T8787-50ML) was added to as-prepared liposome solution, resulting in the gene/drug release. The FAM fluorescence from oligonucleotides (Ex/Em: 494 nm/520 nm) and Dox fluorescence (Ex/Em 485/590 nm) was recorded on a Fluorolog-Tau-3 system (Jobin Yvon-Horiba, US) and compared with the corresponding oligonucleotide and Dox standard curves, respectively. The epotoside amount was determined by measuring the UV absorbance at 285 nm under Cary UV-VIS-NIR absorption spectrophotometer (Varian Incl.) and comparing with the epotoside standard curve.

**$^1$O$_2$ generation tests with light and X-ray triggering**. For light illumination, a 365 nm LED was used to illuminate the samples. 16 μL of SOSG (0.5 mM, Thermo Fisher Scientific Inc, no. S36002) was mixed with 3 mL of liposome suspension and the mixture was then placed in a cuvette, followed by illumination under a 365 nm LED (2.5 mW cm$^{-2}$, irradiation for 10 min). After illumination, the SOSG fluorescence at 525 nm upon 488 nm excitation was recorded using a fluorescence spectrophotometer. For X-ray radiation, a linear accelerator (6 MeV LINAC, Elekta AB, Sweden) was used to deliver different doses (1 Gy, 2 Gy and 4 Gy) to the samples. 96-well plates with 200 μL of liposome suspension and 2 μL of SOSG (0.5 mM) in each well were exposed to X-ray radiation. The irradiation of samples was carried out using 6 MeV X-ray photons from the anterior and posterior directed radiation fields. After irradiation, the SOSG fluorescence was recorded using a microplate reader (PHERAstar FS system, BMG LABTECH, Germany).

**Calcein release assay with light and X-ray irradiation**. Liposomes loaded with calcein were separated from free calcein molecules by using Pall Nanosep® centrifugal devices (Sigma-Aldrich) equilibrated with 10 mM Tris/HCl. They were then activated by light illumination and ionizing radiation, respectively. The experiment process was the same as described in the $^1$O$_2$ generation test, apart from the omission of SOSG. The induced release and subsequent dilution of the calcein previously contained in the liposome s, leading to an increase of calcein fluorescence[65,66]. The calcein fluorescence signal was recorded at 510 nm upon excitation at 485 nm. The percentage of calcein release ($R_c$(%)) at various illumination time points or X-ray dosage was calculated as follows:

$$R_c(\%) = \frac{F_{t(d)} - F_0}{F_{max} - F_0} \times 100\% \qquad (1)$$

where $F_t$ and $F_0$ respectively indicate the fluorescence intensity of calcein at various illumination time points and without illumination. $F_{max}$ refers to the total fluorescence intensity of calcein after the disruption of liposomes by adding 0.1% Triton X-100. For X-ray radiation, $F_d$ is the fluorescence intensity of calcein at various radiation doses, $d$.

**Serum and pH stability studies of PEGylated liposomes**. 200 μL LipoDox samples with and without PEG modification were respectively diluted in PBS (pH 7.4) containing foetal bovine serum (FBS) with different concentrations (0%, 10%, 25 and 50%). All samples were dialyzed in Slide-A-Lyzer MINI dialysis devices (Thermo Fisher Scientific). These devices were then kept in 50 mL centrifuge tubes with 10 mL PBS at 37 °C for 48 h. At various time points (0 h, 2 h, 4 h, 18 h, 24 h and 48 h), an aliquot of PBS was taken for the fluorescence characterisation of the released Dox. The total Dox fluorescence was measured by disrupting liposomes with 0.1% Triton X-100. The percentage of Dox release at various time points was calculated by using the same formula as that applied to the calcein release assays. In our pH-triggered drug release studies, 200 μL Dox-loaded PEGylated liposome suspension was incubated with PBS (containing 10% FBS) with pH respectively adjusted to 7.4 (control), 6.0 and 5.0, followed by the same dialysis procedure and fluorescence measurement described above.

**Cell preparation and ionizing radiation treatment of cells**. Rat PC12 cells, human colon adenocarcinoma HCT 116 cells and normal human colon epithelial cells (CCD 841 CoN) were purchased from the American Type Culture Collection. PC12 cells were cultured in Dulbecco's modified Eagle's medium (DMEM); HCT 116 cells were cultured in McCoy's 5 A (modified) medium; CCD 841 CoN cells were cultured in Eagle's Minimum Essential Medium (EMEM). All culture media were supplemented with 10% foetal bovine serum and 1% antibiotic-antimycotic. The flasks were maintained in a 37 °C incubator with 5% $CO_2$ humidified air. The cells were detached with trypsin and transferred at appropriate dilutions into 96-well plates for cell viability assays or glass-bottom petri dishes for cell imaging. For X-ray radiation experiments, the cells were radiated by using the same accelerator as described in the $^1O_2$ generation test.

**Imaging and analysis of cellular uptake of liposomes**. The PC12 cells ($3 \times 10^4$ mL$^{-1}$) were attached to glass-bottom petri dishes and incubated at 37 °C for 24 h. After removing the culture medium, the cells were incubated with liposome suspension (25 μM) in culture medium supplemented with 10% FBS for 1 h, 4 h and 10 h. The cells were then washed with PBS (1 ×, PH 7.4) three times to remove free liposomes. To assess the uptake of liposome nanoparticles, the cells were fixed with 2.5% paraformaldehyde for 10 min at room temperature, washed twice with PBS (1 ×, PH 7.4) and stained with Hoechst 33342 (5 μg ml$^{-1}$) for 10 min at room temperature before imaging. The cells were imaged using a Leica SP2 confocal laser scanning microscopy system. A violet laser at 405 nm and an argon laser at 496 nm were used for the excitation of VP and FAM-labelled oligonucleotide entrapped inside liposomes, respectively. The imaging of uptake activity of FA-targeted liposomes into HCT 116 cells and CCD 841CoN cells were also conducted as mentioned above.

For quantitative analysis, fluorescently labelled DOTAP (Avanti Polar Lipids, no. 810890 P), was employed, instead of standard DOTAP in order to prepare fluorescent liposomes. PC12 cells ($1 \times 10^4$ mL$^{-1}$) were cultured in petri dishes at 37 °C for 24 h. After removing the old culture medium, 1 mL of a fresh medium containing 10 μL of fluorescently labelled liposomes (0.5 mg mL$^{-1}$) was added to the petri dishes and the cells were incubated at 37 °C for a further 4 h. After incubation, the cells were washed with fresh medium three times to remove free liposomes, detached with trypsin from the petri dishes and counted using a cell counter (Countess II FL automated cell counter from Thermo Scientific). 100 μL NaOH (1 M) and 100 μL Triton X-100 (1% v/v) were subsequently added to 800 μL of cell suspension. The cells were lysed at R.T. for 2 h with constant shaking. After cell lysis, fluorescence (Ex/Em: 460/535 nm) was recorded on a Fluorolog-Tau-3 system and compared with the standard curve of free fluorescent DOTAP solution. A detailed calculation of the number of liposomes per cell is described in Supplementary Note 4.

**Indirect immunofluorescence staining of PAC1R**. The PC12 cells were fixed with 2.5% paraformaldehyde for 10 min and permeabilized with 0.1% Triton X-100 for another 10 min at room temperature, followed by blocking with 5% bovine serum albumin for 30 min. The cells were then incubated with goat anti-PAC1R primary antibody (1:50 dilution, Santa Cruz Biotechnology, no. sc-15964) for 90 min and donkey anti goat IgG secondary antibody (1:100 dilution, Santa Cruz Biotechnology, no. sc-2024) conjugated to FITC for 30 min at room temperature.

**Cytotoxicity assays of X-ray-triggered LipoDox and LipoETP**. The in vitro antitumour effect of X-ray-triggered LipoDox and LipoETP was evaluated using the MTS test. Before treatment, the HCT 116 cells ($2 \times 10^4$ mL$^{-1}$) were grown on 96-well plates in the culture medium with 10% FBS for 24 h. After removing the old medium, the cells were respectively incubated with a series of LipoDox and LipoETP samples diluted in the culture medium with 10% FBS for 4 h. After incubation, the old medium was removed and a fresh medium was added to cells, followed by X-ray radiation with 4 Gy. The cytotoxicity of X-ray-triggered LipoDox and LipoETP on HCT 116 cells at various time points (0 h, 2 h, 4 h and 24 h) was determined by the MTS test (Promega Co., WI, USA, no. G3582) according to manufacturer's instructions and compared with control cells without any treatment. Cell viability was then calculated as a percentage of the absorbance of the untreated control sample. The latter was set to 100%. For comparison purposes, the

viability of cells treated with LipoDox alone was also evaluated in the same experimental conditions.

**Toxicity assays of LipoDox and X-ray on cells and gene**. The PC12, HCT 116 and CCD 841 CoN cells ($1–4 \times 10^4$ mL$^{-1}$) were, respectively, grown on 96-well plates in a culture medium with 10% FBS for 24 h. For liposome and LipoDox treatment experiments, the PC12 cells and CCD 841 CoN cells were, respectively, incubated with different liposome and LipoDox samples for 4 h, followed by incubation in a fresh medium for further 24 h. For the X-ray exposure experiments on cells, all three types of cells were radiated with 4 Gy, followed by incubation in a fresh medium for further 24 and 48 h. Cell viability was assessed by using the same method as described above. For X-ray treatment of pure DNA molecules and mixture of DNA and verteporfin, 50 μL of antisense oligonucleotide solution (10 μg mL$^{-1}$) and 50 μL of mixture solution (10 μg mL$^{-1}$ DNA and 32 μg mL$^{-1}$ verteporfin) was respectively exposed to X-ray radiation with different dosages (1, 2 and 4 Gy). After treatment, the gel electrophoresis was carried out in 1.2 % agarose gel in Tris-acetate-EDTA (TAE) buffer at 95 V for 45 min. The gel was stained with SYBR Safe DNA Gel Stain (Thermo Fisher) and photographed under UV light using a Bio-Rad imaging system.

**In vivo antitumour efficacy by X-ray-triggered drug release**. All procedures were carried out with the approval from Macquarie University Animal Ethics Committee (animal ethics approval No. 2017/001). 6–7 weeks old BALB/c nu/nu female mice (The Animal Resources Centre, Perth, Australia) were injected subcutaneously with $5 \times 10^6$ HCT 116 cells, suspended in 100 μl McCoy's 5 A (modified) medium without FBS, to the flank. Tumours were measured every two days with a caliper and volume (V) was calculated by using the following formula:

$$V = \pi/6 \times L \times W^2 \qquad (2)$$

where L and W are the length (large diameter) and width (short diameter) of the tumour. When tumour volume reached approximately 100 mm³, mice were randomly divided into 4 groups (n = 4 per group) for different treatments: Group A treated PBS via intratumour injection (20 μL); Group B treated with liposome suspension via intratumour injection (20 μL, 10 mg kg$^{-1}$, approximately 10 μM gold nanoparticles and 20 μM VP used for each mouse); Group C treated with X-ray radiation (4 Gy, single fraction) and Group D treated with liposome suspension via intratumour injection (20 μL, 10 mg kg$^{-1}$) and X-ray radiation (4 Gy, single fraction). Mice were then maintained for additional 2 weeks. Body weight and tumour volume were measured every other day. After two weeks, mice were sacrificed and tumours were removed, photographed and fixed with 10% neutral-buffered formalin for histological analysis. Tumour tissues were cryosectioned into serial sections of 6 μm in thickness and stained with haematoxylin and eosin (H&E) following conventional protocol. The histological preparations were examined using an upright research microscope Axio Imager Z2 (Zeiss, Germany) equipped with dry-air EC Plan-Neofluar (5 × /NA0.16; 10 × /NA0.30; 20 × /NA0.50 Ph) and oil-immersion α Plan Apochromat (100×/NA1.46 oil) objectives (Zeiss, Germany). Images were recorded using a digital video camera AxioCam (1388×1040, Zeiss, Germany) in a single-frame and stitching modes.

**Data availability**. The relevant data generated and (or) analysed in the current study are available from the corresponding author upon reasonable request.

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

## Acknowledgements

All TEM images in this work were performed in the Microscopy Unit, Faculty of Science and Engineering at Macquarie University. We thank Dr. Michael Grace, Dr.Vaughan Moutrie and Dr. Daniel Santos from Genesis Cancer Care NSW and Macquarie University Hospital, for helping us with X-ray radiation experiments. We also thank Mr Peter Wieland from the Department of Earth and Planetary Science at Macquarie University for his assistance in ICP-MS measurement. This work is supported by Discovery Early Career Researcher Award scheme (DE130100894), Centre of Excellence scheme (CE140100003) from the Australian Research Council and Sydney Vital Translational Cancer Research Centre. The experimental work was carried out at Macquarie University and Genesis Cancer Care NSW.

## Author contributions

W. D. and E. G. designed this study, conducted experiments and draft the manuscript. W. C. conducted animal experiments, cell lysis experiments and gel electrophoresis. S. C. conducted calculation of SOQY and $^1O_2$ generation enhancement factor under light illumination and the number of $^1O_2$ generated under X-ray radiation. A. G. contributed to histological analysis and interpretation. Z. Z. helped with measurement of SOSG fluorescence intensity and calcein release assays after X-ray radiation. He also contributed to animal ethics application and experiments. A. E. contributed to manuscript preparation.

## Additional information

**Competing interests:** The authors declare no competing interests.

