## [Peer Review File · Nature Communications]

Reviewers' comments:

Reviewer #1 (Remarks to the Author):

Following their initial work on X-ray triggered PDT (Clement et al. Sci. Rep. 2016), the authors have decided to test new applications of X-ray excited metal nanoparticles-photosensitizers for liposomes destabilisation.

Two applications are presented that may broaden the scope: antisense release from liposomes and intracellular antitumoral release.

The data are technically sound and support most of the conclusions of the authors.

However several points have to be taken into account in order to increase the overall quality of the manuscript:

- One control is missing, the measure of the ROS effect on the encapsulated genetic material which is known to be very reacting with ROS.
- Relying onto X-Ray triggering to disrupt liposomes and induce leaking, how do the authors control the intracellular as compared to the extracellular focalisation of the X-rays in order to guarantee an intracellular release of the packed material? This point has to be discussed with respect to physiological triggers.
- While triggered release of antitumoral agents by X-ray make sense due to the potential synergistic effect and the common goal of cells killing, I am more skeptical concerning theirs use for gene silencing due to the potential X-ray secondary effects.
- Figure 2 and the characterisation of the liposomes could find a better place in the supplementary materials. Same for the toxicity assays and figure 8.
- I would suggest to combine figure 3 and 4 because they are providing the reader with complementary informations.

Minor points:

- lack of consistency in tense, for exemple in the legend of the figure 5: « ...scale bar is... », in the legend of figure 6: « ...scale bar was... ».
- figure 2 : « c » missing in scale.
- verteporfin absorbance is not null but « weak » at 525 nm.
- line 166: not « fluorescent lipids » but « fluorescent liposomes ».

Reviewer #2 (Remarks to the Author):

This paper describes a novel releasing approach using a combination of verteporfin and X-ray irradiation through generation of singlet oxygen. Choice of materials is appropriate since the most of them are clinically approved.

However, clinical translation may not be easy since there is a myriad of concerns to be overcome at this stage of studies for radio-, chemo-, and gene therapies. In vivo study is necessary to show the efficacy and in particular, safety, of this novel approach.

For chemotherapy, a liposomal doxorubicin (DOXIL, Janssen Pharmaceutical K.K.) is commercially available, which would demise the novelty of the present approach. Use of doxorubicin could cause myocardial damage when irradiated with therapeutic X-rays. This indicates that combined use of anti-

cancer drug with radiotherapy has not yet been clinically established.

For gene therapy, considering the fact that other delivery techniques such as liposomal delivery (Lipofectamine® 2000, Thermo Fisher Scientific) and viral vectors are available today, clear advantages of the present technique over the others should be addressed. In addition, authors should address the intactness of the genes inside the liposome under enriched conditions of singlet oxygen and water radiolysis products after ionizing irradiation. It may contradict with the concept of triggered release by singlet oxygen, since the main objective of radiotherapy is to cause damages on DNAs.

For radiotherapy, however, combined use of verteporfin and X-ray might contribute to better diagnosis, if the authors could address the minimum dose of X-ray required to release its content, the rate of cargo released, and amount of gold and verteporfin for treatment of specific cancers in animal models.

Furthermore, ionizing X-rays will generate abundant amount of hydroxyl radicals and superoxide anions. It is rarely said that singlet oxygen is a contributor of radiation damage. The authors should refer to therapeutic significance of singlet oxygen, relative to primary water radiolysis products.

Both chemo- and gene therapies require targeting capability to accumulate them specifically on the tumor, because the radiation dose distribution is not exactly focused, as shown in Fig.1. Even with one of the most advanced radiotherapies as IMRT (Intensity Modulated Radio Therapy), a few mm margin is inevitable, where the present liposomes uptaken by the normal cells would harm the normal tissue.

Followings are my comments for specific part of the paper;

L46-50

Importance of spatial targeting is mentioned. This is as important as the triggered release, because non-chargeable X-rays cannot be focused. Compatibility of targeting moiety with verteporfin/gold nanoparticle loading should be discussed.

L97-102

In Fig.2a, it is difficult to identify gold nano clusters. Their locations should be indicated by arrows. Fig.2c shows spectra of liposome complex of interest. The profile of liposomes loaded with VP and gold does not show a surface plasmon resonance peak, which was observed in gold nanoparticle profile. Is the gold loading very low in this case?

L120-122

Enhancement of singlet oxygen generation with gold nanoparticles is speculative, since the distance between them is not controlled. How close should the gold nanoparticles locate to initiate the interaction with verteporfin? For example, the following papers discuss the interaction of porphyrin derivatives with gold nanoparticles within extreme proximity.

Electronic Transport in Porphyrin Supermolecule-Gold Nanoparticle Assemblies

David Conklin†, Sanjini Nanayakkara†, Tae-Hong Park‡, Marie F. Lagadec§, Joshua T. Stecher||, Michael J. Therien||, and Dawn A. Bonnell*†
Nano Lett., 2012, 12 (5), pp 2414–2419
DOI: 10.1021/nl300400a

Ahson J. Shaikh*†‡, Faiz Rabbani†, Tauqir A. Sherazi†, Zafar Iqbal†, Sadullah Mir†, and Sohail A. Shahzad†

Binding Strength of Porphyrin–Gold Nanoparticle Hybrids Based on Number and Type of Linker

Moieties and a Simple Method To Calculate Inner Filter Effects of Gold Nanoparticles Using Fluorescence Spectroscopy

J. Phys. Chem. A, 2015, 119 (7), pp 1108–1116

L147-151

Under X-ray irradiation, the calcein release is increased by 6% with 4Gy dose for verteporfin/gold nanoparticle-loaded liposomes, against verteporfin alone-loaded liposomes. With this increase in release rate, authors should explain a reason to justify 4Gy dose, since this dose is allowed only for therapy.

L179-182

Authors' claim of antisense oligonucleotide release is correct, but intactness of nucleotide should be examined, because ROS generation under 4Gy dose may cause degeneration of DNAs.

L202-205

It would be fair to refer to adverse effects of radiation damages, against a benefit of controlled release with therapeutic X-rays.

Responses to comments

Reviewer A:

Comment A1 - Following their initial work on X-ray triggered PDT, the authors have decided to test new applications of X-ray excited metal nanoparticles-photosensitizers for liposomes destabilisation. Two applications are presented that may broaden the scope: antisense release from liposomes and intracellular antitumoral release. The data are technically sound and support most of the conclusions of the authors. However several points have to be taken into account in order to increase the overall quality of the manuscript:

- One control is missing, the measure of the ROS effect on the encapsulated genetic material which is known to be very reacting with ROS.

Our response: We have now added the toxicity study of ROS generated from VP after X-ray radiation on oligonucleotides to the revised manuscript. The added text now reads:

“For X-ray treatment of pure DNA molecules and mixture of DNA and verteporfin, 50 μ L of antisense oligonucleotide solution (10 μ g/mL) and 50 μ L of mixture solution (10 μ g/mL DNA and 32 μ g/mL verteporfin) was respectively exposed to X-ray radiation with different dosage (1, 2 and 4 Gy). After treatment, the gel electrophoresis was carried out in 1.2 % agarose gel in Tris-acetate-EDTA (TAE) buffer at 95 V for 45 min. The gel was stained with SYBR Safe DNA Gel Stain (Thermo Fisher) and photographed under UV light using a Bio-Rad imaging system.”

“In addition, we also checked the effect of the singlet oxygen on genetic materials by irradiating mixture solution of oligonucleotides and VP with X-ray. As shown in Fig. S9d, there was no clear oligonucleotide damage observed compared with the control.”

Comment A2 - Relying onto X-Ray triggering to disrupt liposomes and induce leaking, how do the authors control the intracellular as compared to the extracellular focalisation of the X-rays in order to guarantee an intracellular release of the packed material? This point has to be discussed with respect to physiological triggers.

Our response: The key physiological trigger for potential liposome disruption is pH. As extracellular and intracellular environment in cancer cells have different pH (5.5-7.0) we evaluated the pH effect on the stability of our liposomes. We incorporated this information in the supplementary information in the revised manuscript. The added text now reads:

“In our pH-triggered drug release studies, 200 μ L Dox-loaded PEGylated liposome suspension was incubated with PBS (containing 10% FBS) with pH respectively adjusted to 7.4 (control), 6.0 and 5.0, followed by the same dialysis procedure and fluorescence measurement described above.”

“Considering that decreased pH is a major feature of tumour tissue and, in principle, it may affect liposome stability and drug release from the liposomes. We also assessed Dox release triggered by different values of pH. These PEGylated liposomes showed a similar Dox release profile at different buffer pH values (7.4, 6.0 and 5.0). The overall amount of released Dox was less than 10% for 48 hr incubation even at pH 5.0.

These findings suggested that the liposome formulation prepared in this study was largely unaffected by the decreased pH. This indicates no stability change of liposomes in the tumour microenvironment before the application of light or X-ray to the tumour site.”

Comment A3 - While triggered release of antitumoral agents by X-ray make sense due to the potential synergistic effect and the common goal of cells killing, I am more skeptical concerning theirs use for gene silencing due to the potential X-ray secondary effects.

Our response: The technology presented in this paper is designed to be used in patients who have already been prescribed radiotherapy. However it is well known that cancer cells become radioresistant as well as chemoresistant. Gene silencing (of cancer genes) represents an additional tool to fight cancer with a different mode of action to radiotherapy and chemotherapy, thus offering a new treatment option.

The key potential secondary effect of X-rays is potential damage of nucleid acids used in gene silencing which would reduce their effectiveness to silence their target genes. In order to clarify this issue we have added the toxicity study of X-ray on oligonucleotides to the revised manuscript. The added text now reads:

“For X-ray treatment of pure DNA molecules and a mixture of DNA and verteporfin, 50 μ L of antisense oligonucleotide solution (10 μ g/mL) and 50 μ L of mixture solution (10 μ g/mL DNA and 32 μ g/mL verteporfin) was respectively exposed to X-ray radiation with different dosage (1, 2 and 4 Gy). After treatment, the gel electrophoresis was carried out in 1.2 % agarose gel in Tris-acetate-EDTA (TAE) buffer at 95 V for 45 min. The gel was stained with SYBR Safe DNA Gel Stain (Thermo Fisher) and photographed under UV light using a Bio-Rad imaging system”.

“With regard to the X-ray effect on genes, the DNA gel electrophoresis did not show an obvious dispersion of DNA bands after X-ray radiation compared to the control, indicating that X-ray radiation with such dosage did not cause noticeable damage to the DNA molecules (Fig. S9d).”

Comment A4 - Figure 2 and the characterisation of the liposomes could find a better place in the supplementary materials. Same for the toxicity assays and figure 8.

Our response: We have moved Fig.2, Fig.8 and Fig.9 to the supplementary material.

Comment A5 - I would suggest to combine figure 3 and 4 because they are providing the reader with complementary informations.

Our response: We have now combined Fig.3 and Fig.4 in the revised manuscript.

Minor points:

- lack of consistency in tense, for example in the legend of the figure 5: « ...scale bar is... », in the legend of figure 6: « ...scale bar was... ».

Our response: We rectified the inconsistency of tense in the description of our scale bar.

- figure 2 : « c » missing in scale.

Our response: We revised Fig.2c in the manuscript (now Fig. S1).

- verteporfin absorbance is not null but « weak » at 525 nm.

Our response: In the revised manuscript, we changed our statement and it now reads:

“It is known that VP molecules have very weak fluorescence at the excitation wavelength of 525 nm.”

- line 166: not « fluorescent lipids » but « fluorescent liposomes ».

Our response: we changed “fluorescent liposomes” to “fluorescently labelled liposomes” in the revised manuscript.

Reviewer B:

Comment B1 - This paper describes a novel releasing approach using a combination of verteporfin and X-ray irradiation through generation of singlet oxygen. Choice of materials is appropriate since the most of them are clinically approved.

However, clinical translation may not be easy since there is a myriad of concerns to be overcome at this stage of studies for radio-, chemo-, and gene therapies. In vivo study is necessary to show the efficacy and in particular, safety, of this novel approach.

Our response: We added new *in vivo* studies in the revised manuscript. We carried out mouse experiments and the results were added to the revised manuscript. The added text is as follows:

“In vivo antitumour efficacy by X-ray triggered drug release

All procedures were carried out with the approval from Macquarie University Animal Ethics Committee (animal ethics approval No. 2017/001). 6-7 weeks old BALB/c nu/nu female mice (The Animal Resources Centre, Perth, Australia) were injected subcutaneously with 5×10^6 HCT 116 cells, suspended in McCoy's 5A (modified) medium without FBS, to the flank. Tumours were measured every two days with a caliper and volume (V) was calculated by using the following formula:

$$V = \pi/6 \times L \times W^2$$

where L and W are the length (large diameter) and width (short diameter) of the tumour. When tumour volume reached approximately 100 mm^3 , mice were randomly divided into 4 groups ($n=4$ per group) for different treatments: Group A treated PBS via intratumour injection (20 μL); Group B treated with liposome suspension via intratumour injection (20 μL , 10mg/kg); Group C treated with X-ray radiation (4 Gy, single fraction) and Group D treated with liposome suspension via intratumour injection (20 μL , 10mg/kg) and X-ray radiation (4 Gy, single fraction). Mice were then maintained for additional 2 weeks. Body weight and tumour volume were measured every other day. After two weeks, mice were sacrificed and tumours were removed, photographed and fixed with 10% neutral-buffered formalin for histological

analysis. Tumour tissues were cryosectioned into serial sections of 6 μm in thickness and stained with haematoxylin and eosin (H&E) following conventional protocol. The histological preparations were examined using an upright research microscope Axio Imager Z2 (Zeiss, Germany) equipped with dry-air EC Plan-Neofluar (5 \times /NA0.16; 10 \times /NA0.30; 20 \times /NA0.50 Ph) and oil-immersion α Plan Apochromat (100 \times /NA1.46 oil) objectives (Zeiss, Germany). Images were recorded using a digital video camera AxioCam (1388 \times 1040, Zeiss, Germany) in a single-frame and stitching modes.

Evaluation on therapeutic effect of X-ray triggered liposomes *in vivo*

To determine the efficacy of X-ray triggered liposomes *in vivo*, we detected their ability to control tumour growth in a xenograft mouse model bearing HCT 116 cells. The sizes of tumours in mice treated with different conditions presented in Fig. 6a. PBS-, liposome- and X-ray-treated tumour respectively increased 3.0-fold, 2.9-fold and 3.4-fold during the whole period (two weeks post treatment), indicating that these treatments failed to delay tumour progression. By contrast, in the group treated with X-ray triggered liposomes the tumour sizes gradually shrunk over this period, with 74% reduction in tumour volume compared to the PBS control group. The size of tumours on mice exposed to different treatments were also photographed and represented in Fig. S10b. This figure shows that tumours in mice treated with X-ray triggered liposomes grew more slowly in comparison with PBS control, X-ray radiation alone and liposomes alone. These findings indicate that such combined treatment can significantly suppress tumour growth, achieving a better therapeutic outcome, compared with individual treatments tested here. In addition, no mortality was observed during 14 days after treatment with X-ray triggered liposomes, and no weight loss of treated mice was observed compared to the control, suggesting that this combined technique was well tolerated by mice under the present conditions (Fig. 6b). Histological analysis was also performed to further verify the tumour response to the treatments. All tumours were found to be localized subcutaneously and surrounded by a thin capsule of connective tissue. No tumour invasion into the capsule tissue was observed. The tumours had a mixed histological structure, with various spatial combinations of viable, paranecrotic and necrotic tumour tissues (Fig.6c). In general, viable tissues were localized mainly at the periphery of the tumours, while the non-viable elements were found more centrally. The mean percentage of necrotised tumour tissues showed statistically significant differences between the studied groups, with the maximal tumour necrosis being achieved when treated with X-ray triggered LipoDox (Fig. 6d). These findings further confirmed that this new strategy can achieve better therapeutic effect compared with individual treatment. More detailed histological analysis for each treatment (PBS-, X-ray-, LipoDox- and X-ray triggered LipoDox-treated) is provided in the supplementary information (Fig. S10a).”

Figure 6. Antitumour activity of X-ray triggered LipoDox in a xenograft model of colorectal cancer. (a and b) Changes of tumours and body weight of mice after various treatments as indicated. Black arrow in Fig. 6a indicates the time of treatment administration. The mean tumour volumes were analysed using the *t* test. * $P < 0.05$, ** $P < 0.01$, *** $P < 0.001$. (c) The structural components of treated tumour (H&E staining). Viable tumour tissues (1) were composed of uniform cells with basophilic (blue) cytoplasm and large roundish hyperchromatic nuclei. The areas of cellular paranecrosis and necrosis (2) were recognized by disorganized groups of tumour cells with eosinophilic (pink) cytoplasm, with and without nuclei, respectively. Arrows indicate congested blood vessels. Note the spatial association between the viable tumour tissue and blood vessels. Scale bar is 50 μm . (d) Morphometric analysis of the effect of the experimental treatment regimens on the structural composition of the xenograft tumours. Relative areas of the viable and non-viable (paranecrotic and necrotic) tumour tissues were measured using ImageJ open source software; the mean tumour necrosis percentage was analysed using the *t* test. * $P < 0.05$, ** $P < 0.01$, *** $P < 0.001$, compared with X-ray triggered LipoDox-treated group.”

The following text was added to the supporting information:

“7. Histological analysis of tumour tissues after each treatment

Fig. S10a demonstrates the images of tumour tissue under different treatment conditions. In the PBS-treated group, the internal region of the tumours are mainly formed by non-viable tumour residuals (about 1/3 of the whole volume of the lesion),

while the outer, part was formed by viable tumour cells. In the LipoDox-treated group, the viable tumour tissue formed a somewhat thinner outer rim (up to 0.5 mm below the capsule), but, in contrast to the PBS-treated group, it protruded towards the inner region as elongated cords and it alternated with the necrotic and paranecrotic sites. The volume of non-viable tumour tissue was about 60-70%. In the X-ray treated group, the difference between the outer and inner regions is less pronounced compared with other groups, and bigger fragments of viable tumour tissue of solid structure are surrounded by non-viable tumour tissue residuals. Finally, in the group treated by X-ray triggered LipoDox, the structure of the tumour resembles that observed in the LipoDox group, but with a significant reduction of the relative volume of viable tumour tissue. In particular, the outer rim of viable tumour tissue is much thinner (only 100-300 μm in thickness), and the amount of the viable tumour tissue spreading into the internal part of the tumour is significantly less than in any other experimental groups. In contrast, the necrotic tumour tissue is visible even in the subcapsular zone. The average volume of necrotic tissue in the tumours is about 80-90%.”

Figure S10 (a) Histological analysis of tumour tissues after various treatments. H & E staining. The scale bar is 100 μm . (b) Photographs of tumours isolated at the endpoint

Comment B2 - For chemotherapy, a liposomal doxorubicin (DOXIL, Janssen Pharmaceutical K.K.) is commercially available, which would demise the novelty of the present approach. Use of doxorubicin could cause myocardial damage when irradiated with therapeutic X-rays. This indicates that combined use of anti-cancer drug with radiotherapy has not yet been clinically established.

Our response: Clinically used doxorubicin is applied in specific doses and the fact that it shows toxicity when used concurrently with radiotherapy is related to that dose. Our approach enables triggered delivery of Dox (and other drugs), when the drug is delivered in short, controlled bursts. This will enable the lowering of the clinical dose and will reopen the question of toxicity during radiation therapy, and the opportunity to use triggered Dox with concurrent X-rays.

In order to fully address reviewer's concern, in addition to Dox, we also loaded another chemotherapy drug, etoposide, into liposomes and checked the cell viability following X-ray triggered drug release from liposomes. The preparation and cell experiments were included in the revised manuscript. It now reads:

“Liposomes incorporating ETP, VP and gold nanoparticles were prepared by thin film hydration with some modifications. Briefly, 100 μL of DOTAP (50 mg/mL in chloroform) was mixed with 54 μL of DOPC (100 mg/mL in chloroform), followed by addition of 6 μL of gold nanoparticle suspension, 7 μL of VP (2.3 mg/mL in DMSO) and 83.5 μL of ETP (Sigma-Aldrich, no. E1383-25MG, 1mg/mL in chloroform and ethanol (1:1, V/V)). After evaporation of organic solvent, the lipid film was hydrated with 1 mL PBS (pH 7.4). The hydration and extrusion procedure was the same as described above. The unloaded etoposide was removed by dialysis in the PBS solution (pH 7.4) with buffer exchange repeated four times.”

“The *in vitro* antitumour effect of X-ray triggered LipoDox and LipoETP was evaluated using the MTS test. Before treatment, the HCT116 cells (2×10^4 /mL) were grown on 96-well plates in the culture medium with 10% FBS for 24 hr. After removing the old medium, the cells were respectively incubated with a series of LipoDox and LipoETP samples diluted in the culture medium with 10% FBS for 4 hr. After incubation, the old medium was removed and a fresh medium was added to cells, followed by X-ray radiation with 4Gy. The cytotoxicity of X-ray triggered LipoDox and LipoETP on HCT116 cells at various time points (0 h, 2 h, 4 h and 24 h) was determined by the MTS test (Promega Co., WI, USA, no. G3582) according to manufacturer's instructions and compared with control cells without any treatment.”

“It should be mentioned that simultaneous chemo- and radiotherapy may result in the development of cardiotoxicity, whose incidence was associated with different factors, including the type of antitumour drugs. Therefore, we evaluated the cell-killing effect of another chemotherapy drug, ETP, in combination with X-ray radiation. ETP is associated with reduced incidence of cardiotoxicity, compared with Dox⁵¹. As shown in Fig. S9c, higher cytotoxicity of LipoETP in HCT116 cells was observed at 24 hours after X-ray radiation of 4 Gy, compared with LipoETP alone.”

Comment B3 - For gene therapy, considering the fact that other delivery techniques such as liposomal delivery (Lipofectamine® 2000 , Thermo Fisher Scientific) and viral vectors are available today, clear advantages of the present technique over the others should be addressed.

Our response: We added some drawbacks of viral carriers and Lipofectamine 2000, compared with our liposome delivery system in the revised manuscript. The added text now reads:

“Although viral carriers have been traditionally used as a gene/drug delivery method^{1, 2}, their application is hindered by a range of limitations including immunogenicity, limited size of transgenic materials, packaging difficulties and the risk of recombination³. Furthermore, viral carriers do not offer any temporal control over transfection which, once introduced, can not be deliberately stopped⁴. To overcome these limitations, synthetic nanomaterial-based systems have been extensively studied and developed.”

“Conventional liposomes, for example, commercial lipofectamine 2000, are unsuitable for the on-demand content release, which limits their therapeutic applications, although they show high efficiency of delivery.”

Comment B4 - In addition, authors should address the intactness of the genes inside the liposome under enriched conditions of singlet oxygen and water radiolysis products after ionizing irradiation. It may contradict with the concept of triggered release by singlet oxygen, since the main objective of radiotherapy is to cause damages on DNAs.

Our response: We added the toxicity study of water radiolysis products and singlet oxygen generated from VP after X-ray radiation on oligonucleotides. The added text now reads now reads:

“For X-ray treatment of pure DNA molecules and mixture of DNA and verteporfin, 50 μ L of antisense oligonucleotide solution (10 μ g/mL) and mixture solution of DNA and verteporfin (10 μ g/mL DNA and 32 μ g/mL verteporfin) was exposed to X-ray radiation with different dosage (1, 2 and 4 Gy). After treatment, the gel electrophoresis was carried out in 1.2 % agarose gel in Tris-acetate-EDTA (TAE) buffer at 95 V for 45 min. The gel was stained with SYBR Safe DNA Gel Stain (Thermo Fisher) and photographed under UV light using a Bio-Rad imaging system.”

“With regard to the X-ray effect on the genetic material, the DNA gel electrophoresis did not show obvious dispersion of DNA bands after X-ray irradiation compared to the control, indicating that X-ray radiation at the applied dosage did not cause obvious damage to the DNA molecules (Fig. S9d).”

“In addition, we also checked the effect of the singlet oxygen on genetic materials by irradiating a mixture of oligonucleotides and VP with X-ray. As shown in Fig. S9d, no clear oligonucleotide damage was observed compared with the control.”

Comment B5 - For radiotherapy, however, combined use of verteporfin and X-ray might contribute to better diagnosis, if the authors could address the minimum dose of X-ray required to release its content, the rate of cargo released, and amount of gold and verteporfin for treatment of specific cancers in animal models.

Our response: We assessed the different parameters for the optimal treatment effect by combining *in vitro* and *in vivo* work. Three different doses (1, 2 and 4 Gy) of X-ray radiation were used *in vitro* work and 4 Gy was further used in a xenograft model bearing colorectal cancer. Based on the observed therapeutic effects and safety, the minimal dose was 4 Gy in the current treatment condition. The rate of Dox release is

determined by calcein release assays and equivalent to the amount of released calcein tested in the solution provided in Fig.2. In addition, we calculated the amount of gold and VP based on the liposome dose used in the animal work (10 mg/kg, 20 μ L each mouse), with approximately 10 μ M gold nanoparticles and 20 μ M VP being used for treatment in a colorectal cancer mouse model.

We added these statements to the revised manuscript and it now reads:

“20 μ L, 10mg/kg, approximately 10 μ M gold nanoparticles and 20 μ M VP used for each mouse”

“Based on the in vitro work, 4 Gy was chosen for radiation on mice.”

Comment B6 - Furthermore, ionizing X-rays will generate abundant amount of hydroxyl radicals and superoxide anions. It is rarely said that singlet oxygen is a contributor of radiation damage. The authors should refer to therapeutic significance of singlet oxygen, relative to primary water radiolysis products.

Our response: In this study, singlet oxygen or other ROS generated by VP under X-ray radiation were used to destabilise the lipid bilayer, so they are not used for cell damage. It is the chemo drug released from liposomes which is primarily responsible for cancer cell death after X-ray radiation. To respond to this comment we added references to therapeutic effects and side effects of singlet oxygen and water radiolysis products to the revised manuscript. The revised text now reads:

“Singlet oxygen is the primary cytotoxic agent responsible for photobiological activity involved in the PDT technique. It can damage cells by reacting with many biomolecules, including amino acids, nucleic acids and unsaturated fatty acids that have double bonds as well as sulphur-containing amino acids^{5,6}. The short lifetime of singlet oxygen prevents it from travelling larger distances, therefore it mainly causes localised⁷, nanometre scale damage, near the photosensitizer molecule where it was generated. In this study singlet oxygen generated from VP loaded in a lipid bilayer mainly destabilises the unsaturated lipid and consequently induces drug release. This reaction with lipids consumes singlet oxygen radicals⁸. Therefore, adverse effect of singlet oxygen on oligonucleotides will be minimised.”

Comment B7 - Both chemo- and gene therapies require targeting capability to accumulate them specifically on the tumor, because the radiation dose distribution is not exactly focused, as shown in Fig.1. Even with one of the most advanced radiotherapies as IMRT (Intensity Modulated Radio Therapy), a few mm margin is inevitable, where the present liposomes uptaken by the normal cells would harm the normal tissue.

Our response: We functionalised our liposome nanoparticles by conjugation with PEG and folic acid and evaluated the specific targeting property and serum stability of these nanoparticles. The related content was included in the revised manuscript, and it now reads:

“Preparation of folate-conjugated liposomes

Folate-conjugated liposomes were prepared by postinsertion of DSPE-PEG2000-Folate micelles into preformed liposomes with slight modifications. In brief, 1 mg DSPE-PEG2000-folate (Avanti Polar Lipids, no. 880124) was dissolved in 320 μ L DMSO, followed by hydration with 3.1 mL of distilled water, producing 100 μ M micelle suspension. The suspension was then dialyzed three times in a 10000 MWCO dialysis tubing against 1 L water to remove DMSO. After this, 40 μ L of micelles were added to 1 mL of the preformed liposome suspension in ammonium sulphate (250 mM) and heated at 60 °C for 1 hour to produce folate-tethered liposomes. Leaked ammonium sulphate and unincorporated micelles were removed by dialysis. To determine the folate content conjugated with liposomes, bare liposomes was used in conjugation procedure instead of VP-loaded liposomes. After preparation, the folate amount was determined by measuring the UV absorbance at 285 nm after lysing liposomes with 0.1% Triton X-100 and comparing with a standard curve of folic acid with the known concentration.

Cellular uptake activity of folate-conjugated liposomes for colorectal cancer cells

The folate receptor (FR) is significantly expressed in many types of cancer cells while its expression in most normal tissues is generally low⁹. Folic acid (FA) has a very high affinity for the FRs with a minimal effect on its binding ability even after conjugation with other nanomaterials. Therefore FA can significantly enhance the capability of nanoparticle-based delivery systems to target cancer cells^{10, 11}. In this study, we modified the liposome surface with folate and determined the average number of the folate molecules per liposome based on the total amount of folate and liposomes in the sample, which is estimated to be approximately 480. To evaluate the targeting specificity of the folate-conjugated liposomes to tumour cells, the uptake activity of liposomes by colorectal cancer HCT 116 cells, was compared to the uptake by normal human colonic epithelium CCD 841 CoN cells. As shown in Fig. 4, cancer cells treated with folate-conjugated liposome nanoparticles clearly exhibited the red signal from VP in the cytoplasm after 1 h incubation. By contrast, the level of liposome uptake by CCD 841 CoN cells is shown to be fairly low under the same experimental conditions. These results indicated that FA induced specific binding to the folate receptor expressed on HCT 116 cell surface, resulting in a much higher internalization rate of targeted liposomes, compared to the normal CCD 841 cells.”

We also added the following section to the supporting information

“Serum and pH stability studies of PEGylated liposomes

For serum stability studies, the cumulative percentage of Dox released from liposomes with and without PEG modification is shown in Fig. S5a and S5b. Different amounts of Dox were released from conventional liposomes during 48 hr incubation, with the total amount being more than 30% and 50% at 48 hr when incubated in PBS with 10% and 50% FBS (Fig. S5a). However, the Dox release profile shown in Fig. S5b indicates that the release rates were reduced in the PEGylated liposomes, compared with liposomes without PEGylation. Liposomes still retained more than 90% and 80% of their initial drug content at 48 hr incubated in PBS with 10% and 50% FBS, indicating that PEG chains on the liposome surface would contribute to improved its stability in the blood circulation. Considering that decreased pH is a major feature of tumour tissue and it is likely to affect drug release from liposomes, we also assessed Dox release triggered by pH with different values. These PEGylated liposomes showed a similar Dox release profile at different buffer

pH values (7.4, 6.0 and 5.0). The overall amount of released Dox was less than 10% for 48 hr incubation even at pH 5.0 (Fig.S5c). These findings suggest that liposome formulation prepared in this study was largely unaffected by decreased pH value, which ensures the stability of liposomes in the tumour microenvironment before application of light or X-ray to a tumour site.

Followings are my comments for specific part of the paper;

Comment B8 - L46-50

Importance of spatial targeting is mentioned. This is as important as the triggered release, because non-chargeable X-rays cannot be focused. Compatibility of targeting moiety with verteporfin/gold nanoparticle loading should be discussed.

Our response: In clinical practice X-ray radiation dose can be precisely delivered to the target tumour site with the minimal damage on the normal tissue with the assistance of advanced therapy planning software. To further enhance the tumour-focusing capability of this combined technique (X-ray triggered liposomes), we modified the liposome surface by attaching another lipid component containing targeting moiety to prepared liposomes. To demonstrate this property, we conjugated liposomes with folic acid (targeting molecules for cancer cells) and PEG polymer by this technique. The revised text can be found in our response to *Comment B7*.

Comment B9 -L97-102

In Fig.2a, it is difficult to identify gold nano clusters. Their locations should be indicated by arrows.

Our response: We added arrows to indicate the gold nanoclusters in the revised manuscript (Fig.S1). We also added the TEM image of pure liposomes without gold loading to the revised manuscript. Compared to pure liposomes, gold nanoparticles loaded into the liposomal bilayer can be easily observed (Fig.S1)

Fig.2c shows spectra of liposome complex of interest. The profile of liposomes loaded with VP and gold does not show a surface plasmon resonance peak, which was observed in gold nanoparticle profile. Is the gold loading very low in this case?

Our response: Yes, the gold loading was fairly low in our liposomes. The surface plasmon resonance peak of gold nanoparticles around 485 nm was observed in the inset of Fig. S1.

Comment B10 - L120-122

Enhancement of singlet oxygen generation with gold nanoparticles is speculative, since the distance between them is not controlled. How close should the gold nanoparticles locate to initiate the interaction with verteporfin? For example, the following papers discuss the interaction of porphyrin derivatives with gold nanoparticles within extreme proximity.

Electronic Transport in Porphyrin Supermolecule-Gold Nanoparticle Assemblies

David Conklin†, Sanjini Nanayakkara†, Tae-Hong Park‡, Marie F. Lagadec§,

Joshua T. Stecher||, Michael J. Therien||, and Dawn A. Bonnell†*

Nano Lett., 2012, 12 (5), pp 2414–2419

DOI: 10.1021/nl300400a

Ahson J. Shaikh†‡, Faiz Rabbani†, Tauqir A. Sherazi†, Zafar Iqbal†, Sadullah Mir†, and Sohail A. Shahzad†*

J. Phys. Chem. A, 2015, 119 (7), pp 1108–1116

Our response: We added the relevant statements in the revised manuscript, and it now reads:

“However such enhancement was dependent on one of experimental factors, the distance between gold and photosensitisers. In this study the distance between gold and VP was not controllable under the current condition because both of them was randomly loaded in the liposomal bilayer, with some molecules less than optimally placed in term of the distance for optimal enhancement of singlet oxygen. In particular, the interaction between gold and photosensitisers would not contribute to the singlet oxygen generation when they are within extreme proximity^{12,13}. This may partially contribute to limited enhancement observed in this study.”

We cited the references provided by the reviewer in the revised manuscript (ref 40 and 41).

Comment B11 - L147-151

Under X-ray irradiation, the calcein release is increased by 6% with 4Gy dose for verteporfin/gold nanoparticle-loaded liposomes, against verteporfin alone-loaded liposomes. With this increase in release rate, authors should explain a reason to justify 4Gy dose, since this dose is allowed only for therapy.

Our response: In our calcein release study, although its release percentage was enhanced with dose increase, the rate reached a plateau at 4Gy. We also checked cell viability and the mouse survival after irradiation at 4 Gy, and we found that the cell survival was not affected after 24 hours and the mouse body weight was not reduced within two weeks. These findings indicated that 4Gy was appropriate for X-ray-triggered drug release both in vitro and in vivo applications.

Comment B12 - L179-182

Authors' claim of antisense oligonucleotide release is correct, but intactness of nucleotide should be examined, because ROS generation under 4Gy dose may cause degeneration of DNAs.

Our response: We added a toxicity study of singlet oxygen generated from VP after X-ray radiation on oligonucleotides. The added text now reads:

“For X-ray treatment of pure DNA molecules and mixture of DNA and verteporfin, 50 μ L of antisense oligonucleotide solution (10 μ g/mL) and mixture solution of DNA and verteporfin (10 μ g/mL DNA and 32 μ g/mL verteporfin) was exposed to X-ray radiation with different dosage (1, 2 and 4 Gy). After treatment, the gel electrophoresis was carried out in 1.2 % agarose gel in Tris-acetate-EDTA (TAE) buffer at 95 V for 45 min. The gel was stained with SYBR Safe DNA Gel Stain (Thermo Fisher) and photographed under UV light using a Bio-Rad imaging system.”

“In addition, we also checked the effect of the singlet oxygen on genetic materials by irradiating mixture solution of oligonucleotides and VP with X-ray. As shown in Fig. S9d, there was no clear oligonucleotide damage observed compared with the control.”

Comment B13- L202-205

It would be fair to refer to adverse effects of radiation damages, against a benefit of controlled release with therapeutic X-rays.

Our response: We added references to adverse effects of X-ray radiation to the revised manuscript. The modified text now reads:

“It is well known that radiolysis of water molecules as a result of X-ray radiation damages DNA molecules by producing toxic radicals. Although cells attempt to repair the damage, complete repair may not be possible at higher doses⁵⁵. The surviving cells may suffer residual DNA damage, potentially contributing to adverse long term health effects. In this study we particularly assessed the toxicity of X-ray on both cultured cells and genetic materials.”

“X-rays and other forms of ionizing radiation clinically used to diagnose and treat various medical conditions are known to contribute to DNA mutations at the cellular level. This may lead to increased health problems, compared with the effects of light.

However X-rays with the suitable energy can much more easily penetrate through tissues compared to light, activating gene/drug release in deep tissues once the X-ray triggered liposomes reach their target.”

References:

1. Thomas, C.E., Ehrhardt, A. & Kay, M.A. Progress and problems with the use of viral vectors for gene therapy. *Nature reviews. Genetics* **4**, 346 (2003).
2. Zhang, Y., Satterlee, A. & Huang, L. In vivo gene delivery by nonviral vectors: overcoming hurdles? *Molecular therapy* **20**, 1298-1304 (2012).
3. Luo, D. & Saltzman, W.M. Synthetic DNA delivery systems. *Nat Biotechnol* **18**, 33-37 (2000).
4. Liu, D., Yang, F., Xiong, F. & Gu, N. The smart drug delivery system and its clinical potential. *Theranostics* **6**, 1306 (2016).
5. Niedre, M., Patterson, M.S. & Wilson, B.C. Direct Near-infrared Luminescence Detection of Singlet Oxygen Generated by Photodynamic Therapy in Cells In Vitro and Tissues In Vivo. *Photochemistry and photobiology* **75**, 382-391 (2002).
6. DeRosa, M.C. & Crutchley, R.J. Photosensitized singlet oxygen and its applications. *Coordination Chemistry Reviews* **233**, 351-371 (2002).
7. Krieger-Liszkay, A. Singlet oxygen production in photosynthesis. *Journal of experimental botany* **56**, 337-346 (2005).
8. Redmond, R.W. & Kochevar, I.E. Spatially resolved cellular responses to singlet oxygen. *Photochemistry and photobiology* **82**, 1178-1186 (2006).
9. Parker, N. et al. Folate receptor expression in carcinomas and normal tissues determined by a quantitative radioligand binding assay. *Analytical biochemistry* **338**, 284-293 (2005).
10. Hong, E.J., Choi, D.G. & Shim, M.S. Targeted and effective photodynamic therapy for cancer using functionalized nanomaterials. *Acta Pharmaceutica Sinica B* **6**, 297-307 (2016).
11. Zhang, S., Lu, C., Zhang, X., Li, J. & Jiang, H. Targeted delivery of etoposide to cancer cells by folate-modified nanostructured lipid drug delivery system. *Drug delivery* **23**, 1838-1845 (2016).
12. Conklin, D. et al. Electronic transport in porphyrin supermolecule-gold nanoparticle assemblies. *Nano letters* **12**, 2414-2419 (2012).
13. Shaikh, A.J. et al. Binding Strength of Porphyrin– Gold Nanoparticle Hybrids Based on Number and Type of Linker Moieties and a Simple Method To Calculate Inner Filter Effects of Gold Nanoparticles Using Fluorescence Spectroscopy. *J. Phys. Chem. A* **119**, 1108-1116 (2015).

REVIEWERS' COMMENTS:

Reviewer #1 (Remarks to the Author):

The authors have now answered all the points raised by both reviewers with strong arguments and a lot of complementary experiments which substantiate their claims. With those new experiments and figures the article is now opening new applications possibilities to X-Ray radiation triggering in pharmacological sciences and could be published with very few modifications.

I will just recommend to change histograms for boxplots in figures 5, 6, S8 and S9 for better comparison.

Reviewer #1 (Remarks to the Author):

The authors have now answered all the points raised by both reviewers with strong arguments and a lot of complementary experiments which substantiate their claims. With those new experiments and figures the article is now opening new applications possibilities to X-Ray radiation triggering in pharmacological sciences and could be published with very few modifications.

I will just recommend to change histograms for boxplots in figures 5, 6, S8 and S9 for better comparison.

Our response: We have changed the histograms for boxplots in Fig.5, 6, S8 and S9.